# Variabilities in N2 and E Gene Concentrations in a SARS-CoV-2 Wastewater Multiplex Assay

**DOI:** 10.3390/microorganisms13081862

**Published:** 2025-08-09

**Authors:** Ashley Green, Aiswarya Rani Pappu, Melanie Oakes, Suzanne Sandmeyer, Matthew Hileman, Sunny Jiang

**Affiliations:** 1Department of Civil and Environmental Engineering, University of California Irvine, Irvine, CA 92697, USA; ajgreen2@uci.edu (A.G.); pappua@uci.edu (A.R.P.); 2Department of Biological Chemistry, University of California Irvine, Irvine, CA 92697, USA; mloakes@uci.edu (M.O.); sbsandme@uci.edu (S.S.); 3Joe C. Went School of Population and Public Health, University of California Irvine, Irvine, CA 92697, USA; hilemanm@uci.edu; 4Department of Ecology and Evolutionary Biology, University of California Irvine, Irvine, CA 92697, USA

**Keywords:** wastewater-based surveillance, manhole, COVID-19, RT-ddPCR, primer, recovery efficiency

## Abstract

Wastewater can serve as both a source of pathogens that pose risks to human health and a valuable resource for tracking and predicting disease prevalence through wastewater-based surveillance (WBS). In WBS for SARS-CoV-2, both nucleocapsid-specific (N1 and N2) and the envelope (E) genes are common targets for primer design, but ambiguity remains regarding differences in results depending on the gene target chosen. This study investigated how and why two SARS-CoV-2 gene targets (N2 and E) varied when analyzed in a multiplex RT-ddPCR assay for a COVID-19 wastewater monitoring study. From December 2021 to June 2022, over 700 raw wastewater samples were collected from thirteen manholes in the University of California, Irvine sewer system. Murine hepatitis virus (MHV) was used as a matrix recovery and process control in the triplex RT-ddPCR assay. Water quality tests (TSS, COD, pH, turbidity and NH_3_-N) were performed on all samples. Analyses showed that in over 10% of samples, the E gene concentration exceeded N2 by more than one order of magnitude. To evaluate matrix effects on amplification efficiency for N2 and E genes, multiple regression analysis was performed to explore whether water quality variables and MHV recovery efficiency could predict variance in gene concentrations, but no clear relationship was identified. However, viral recovery, as indicated by MHV recovery efficiency, was negatively impacted in samples with higher TSS and COD, suggesting PCR inhibition. These findings contribute to methodological standardization efforts in WBS and emphasize the importance of primer selection for large-scale monitoring.

## 1. Introduction

Wastewater is an important source of waterborne disease transmission but also a new tool for diagnosing the spread of infectious disease in the community [1,2]. The latter is well demonstrated in the intense effort of wastewater-based surveillance (WBS) for SARS-CoV-2 in the past five years [3,4,5,6,7,8]. Accurate and sensitive detection of pathogens in wastewater is critical for both water quality monitoring and tracking the spread of disease through wastewater surveillance. The advancements of molecular technology, specifically quantitative PCR assays, have significantly improved the sensitivity, specificity, and accuracy of pathogen monitoring in water. These methods detect specific gene targets unique to a pathogen through amplification. However, monitoring results may be significantly influenced by the selection of PCR gene targets, and therefore, the interpretation of the outcomes. While PCR primer design and gene target selection have improved over the past three decades, large-scale quantitative analyses of variability across different gene targets for the same pathogen appear to be lacking.

The onset of the COVID-19 pandemic showed systemic and global public health unpreparedness, and this shock to public health systems caused typical clinical disease surveillance tools to be exasperated. Public health entities had to switch from passive surveillance methods, characterized by long-term routine reporting, to an active surveillance approach to keep up with the rapid spread of COVID-19. Uniquely, as early as April 2020, this active surveillance approach included wastewater-based monitoring of SARS-CoV-2, the virus that causes COVID-19 [9,10]. The scale of WBS of SARS-CoV-2 is unprecedented—with thousands of studies and counting being implemented since the first published US COVID-19 findings, according to the Centers for Disease Control and Prevention (CDC) [11,12].

SARS-CoV-2 is a single-stranded, RNA-enveloped virus that is shed in human feces during COVID-19 infection and persistent in wastewater. There are several studies discussing the different methods of detection and quantification of the SARS-CoV-2 virus, and a crucial component of this is deciding which SARS-CoV-2 gene(s) to target for amplification and the primers to use [5,7,13,14,15,16,17,18].

The SARS-CoV-2 genome is around 30 kb in length, contains 14 open reading frames (ORFs), and encodes 29 viral proteins [19]. Of these viral proteins, there are four crucial structural proteins—spike (S), membrane (M), envelope (E), and nucleocapsid (N)— that are all responsible for virion assembly [20]. Many gene targets associated with these structural proteins are used in WBS for SARS-CoV-2, but there is ambiguity surrounding the differences in the results based on the chosen gene target.

Due to the CDC publishing open-source primer/probe sequences for detection of the SARS-CoV-2 N gene and the World Health Organization (WHO) publishing E gene primer/probes, these genes are common targets used for WBS projects. Thus, it is common for duplex or even triplex (multiplex) assays to be developed to target multiple SARS-CoV-2 genes or gene regions at once to ensure satisfactory amplification of the viral target and detection in complex wastewater matrices [8,21]. Some studies have shown the viability of these SARS-CoV-2 multiplexing PCR assays and even reported results of varying gene concentrations to compare to clinical cases [21,22].

The N gene is an effective target due to its relative abundance during replication. This abundance could enhance the diagnostic sensitivity of PCR, which is useful for analyzing complex environmental matrices prone to PCR inhibition, such as wastewater [6,23]. Three target gene regions, N1, N2, and N3, were established by the CDC, and N1 and N2 were explicitly designed to target SARS-CoV-2 versus N3, which was designed to detect all clade 2 and 3 viruses within the Sarbecovirus subgenus [6]. The E gene is a highly conserved structural protein region, meaning it is less prone to mutations of its genetic sequence. The E gene primer sequence, published by the WHO, was initially used for SARS-CoV-2 detection by researchers in Germany [24].

With WBS standards slowly starting to homogenize in the US through the development of public-facing reporting dashboards for wastewater viral concentrations, the genes that are reported more frequently are the reliably detectable and CDC-published N gene targets. Thus, when reporting of wastewater SARS-CoV-2 concentration data to the University of California, Irvine (UCI) Provost’s office via a public-facing dashboard for an on-campus COVID-19 study, the N2 gene was reported since it is widely used in the literature and supported by the CDC and other US public health entities [25]. However, when looking back at this expansive dataset, variabilities in the detection of the N2 and E genes led us to question the selection of this target choice. There have been studies that reported their results of the N2 and E gene targets [13,22,26,27,28,29,30,31,32] and many of these studies acknowledge the differences in amplification of the N2 vs. E gene, noting that the E gene was less stable than the N2 gene fragment [22] or even that E gene levels were consistently below the limit of detection, despite the N2 gene being amplified [13,29]. Although these observations of variability were noted in these studies, they lacked a systematic analysis of the variability of genes targets and the possible causes of PCR inhibition in the wastewater samples by wastewater composition.

This study seeks to determine whether single-target reporting in wastewater monitoring efforts for SARS-CoV-2 can sufficiently represent the disease state in a community under surveillance. By conducting a comparative study of SARS-CoV-2 N2 and E gene concentrations, the recovery control efficiency of the murine hepatitis coronavirus (MHV), and water quality parameters (chemical oxygen demand (COD), total suspended solids (TSS), ammonia nitrogen (NH_3_-N), pH, and turbidity), this study hopes to better understand 1) the best approach for reporting SARS-CoV-2 concentrations and 2) which physio-chemical characteristics of wastewater may have contributed to the over- or under-amplification of the gene targets during RT-ddPCR. Through this analysis, the study contributes to the understanding of strategic primer and gene target selection in SARS-CoV-2 and sheds light on the influences of gene target selection in the monitoring of waterborne pathogens in wastewater.

## 2. Materials and Methods

### 2.1. Wastewater Sampling and Water Quality Analysis

Wastewater samples (24 h composite) were collected from sewer manholes at UCI campus residential communities from December 2021 to June 2022. Thirteen manhole sites were selected, ensuring wastewater monitoring coverage of nearly all undergraduate and graduate housing facilities. Detailed information on sampling location, collection methods, and frequencies are presented in our previous wastewater epidemiological study [33]. Wastewater aliquots were tested for water quality indicators including COD, TSS, NH_3_-N, pH, and turbidity. These water quality parameters were chosen due to their wide use in the wastewater engineering industry, ability to be pseudo-indicators of human waste, and possible correlation to virus levels due to enveloped viral particles’ affinity for solids in wastewater [34,35,36,37].

Wastewater aliquots were stored at −80 °C upon arrival at the lab for 2 to 12 months, until all the samples were processed. Before testing, frozen samples were thawed at 4 °C slowly for 24 to 48 h to minimize interference with sample integrity, and subsequently analyzed for COD, TSS, NH_3_-N, pH, and turbidity. Hach testing kits and the DR-900 Colorimeter (Catalog #9385100, Hach, Loveland, CO, USA) were used to quantify the COD and NH_3_-N content (Catalog #2125915 and #2606945 respectively, Hach, Loveland, CO, USA). TSS was measured according to the ASTM D5907 Standard Methods for Non-Filterable Matter, with slight modifications detailed in the Appendix A [38]. All samples were run in duplicates, alongside blank and positive control(s) if needed for normalization with the testing instrument. Over 2700 water quality tests were conducted on the wastewater samples collected and stored during this study.

### 2.2. Quantification of N2 and E Gene Regions of SARS-CoV-2

Freshly collected wastewater samples were spiked with a known concentration of murine hepatitis virus (MHV) upon arrival at the lab. MHV was used as the matrix recovery control to evaluate virus loss/inhibition during sample concentration, nucleic acid extraction, and RT-ddPCR amplification. The spiked samples were concentrated on KingFisher 24 deep-well plates (Catalog #95040470, Thermo Fisher, Waltham, MA, USA) using Nanotrap Magnetic Virus Particles (Catalog #44202, Ceres Nano, Manassas, VA, USA). The viral concentrates were then lysed using the MagMAX™ Microbiome Lysis Solution and the nucleic acid purifications were performed using the MagMAX™ Microbiome Ultra Nucleic Acid Isolation Kit (Catalog #A42357, Thermo Fisher). The details of sample concentration and nucleic acid extraction and purification are reported in Pappu et al. [33].

Quantification of N2 and E gene of SARS-CoV-2 was performed on the same day of nucleic acid purification on the Bio-Rad QX200 RT-ddPCR System (Catalog #1864003, Bio-Rad, Hercules, CA, USA) using PREvalence RT-ddPCR SARS-CoV-2 Wastewater Quantification Kit (#12015402). This kit is a triplex RT-ddPCR assay, which includes three sets of primers targeting N2, E, and MHV gene, respectively. The primer validation and QA/QC were performed according to manufacturer protocol. The limit of detection for N2 gene, E gene, and MHV are 5, 2, and 30 genome copies (GC)/µL, respectively. Samples that did not meet the control standards were reprocessed in accordance with the manufacturers’ QA/QC recommendations (Bio-Rad). Samples with MHV recovery control percentage exceeding 100%, indicating control seeding error, sample analysis error, or indigenous MHV, were excluded from analysis. A detailed description of the assay protocols is provided in Pappu et al. [33]. Over 700 data points for SARS-CoV-2 N2, E gene and MHV recovery were collected over the study period. Each data point represents the average from triplicate assays.

### 2.3. Data Analysis

Data analyses were performed using MS Excel program, OriginPro 2025b (OriginLab Corporation, Northampton, MA, USA), and RStudio 2025.05.1+513 (Posit Software, PBC, Boston, MA, USA). Mann–Whitney U Test, a non-parametric statistical test, was performed to compare the MHV recovery efficiency between samples categorized into different water quality groups. Multiple regression tests were performed in R, and all other statistical tests were performed in OriginPro. A *p*-value of <0.05 was considered significant.

## 3. Results

### 3.1. N2 Gene Underestimates SARS-CoV-2 Concentration in Wastewater

A correlation analysis of N2 and E gene of SARS-CoV-2 revealed a strong, significant positive linear correlation (r_s_ = 0.89), suggesting the two gene targets are in general agreement among the 700 data points over the 6-month temporal period (Figure 1a). This result supports the approach of using N2 gene to reflect the community spread of SARS-CoV-2 as shown in our previous wastewater-based epidemiological study on campus [33]. However, a closer comparison of the two gene concentrations revealed discrepancies throughout the dataset (Figure 1b–d). The two gene concentrations agreed within one order of magnitude (1-log) for 86% of samples. However, in 14% of samples, the E gene concentration was more than one order of magnitude (1-log) greater than N2 (Figure 1b). The higher concentrations of E gene were more frequently observed during December and between February and March, when the SARS-CoV-2 concentrations were lower in the wastewater and clinical case reports were also lower in the community [33]. As revealed in Figure 1b, there is only one instance of the N2 gene concentration having a higher gene concentration than the E gene by over a 1-log difference (1.21-log difference on 31 January 2022 at site COM_G2). The frequent negative log differences (logN2–logE) in Figure 1b shows that the average E gene concentrations exceeded N2 concentrations at a higher frequency. Therefore, this result suggests that using N2 gene alone as the SARS-CoV-2 marker may underestimate or miss the detection of SARS-CoV-2 when the case numbers and wastewater concentrations are low.

The relationship between SARS-CoV-2 concentrations and the discrepancies between the two genes was analyzed further using data collected from one of the manhole sites, which is also a confluence point (COM_H). This site serves a population of over 2500 students and is centrally located in the middle of campus to serve an expansive area of undergraduate and graduate student housing communities and receives consistent wastewater flows (Appendix A). At this site, the average N2 concentrations ranged from 0 to 1.33 × 10^4^ genome copies per milliliter (copies/mL), with an average of 7.74 × 10^2^ copies/mL, while the E gene ranged from 0 to 1.72 × 10^4^ copies/mL with an average of 9.08 × 10^2^ copies/mL (Figure 2). The average E gene concentration exceeded the N2 gene concentration by over 57% of the time (Figure 2). The exceeding concentrations were more frequently observed when the SARS-CoV-2 concentrations were below 100 copies/mL (Figure 2). In less than 13% of the time, higher N2 gene concentrations were observed when the SARS-CoV-2 concentrations were above 1000 copies/mL for both gene targets, especially in May and June, when clinical cases were higher [33]. This analysis over the expansive dataset from 13 collection sites and a 6-month period confirms that the N2 gene is a more efficient target when the wastewater viral concentration is high, while the E gene performs better when the viral concentration is low.

From a public health perspective, missing a positive detection is more consequential than measuring the exact virus concentration, as it could delay recognition of a new pandemic wave and slow the public health response. Considering this, a more detailed assessment of SARS-CoV-2 data was performed for every individual assay in triplicates where the N2 gene was detected in none of the three assays, while the E gene target was detected at adequate levels in all triplicates for the same sample. This analysis intends to reveal misdiagnosis of positive samples due to the low concentration of viruses. As shown in Figure 3, there were 188 instances where the N2 gene was not detected, while E gene was detected in all triplicates. In contrast, there were 71 instances where E gene was not detected, while a successful amplification of N2 gene target was observed in the same sample. Figure 3 shows the frequency distribution of E gene concentrations, when N2 was not detectable (concentration = 0 GC/mL). Among the 188 instances, the majority (52.1%) of instances showed amplification between 10 and 100 copies/mL, indicating low level of COVID-19 cases among the community. The other 34.6% of instances revealed SARS-CoV-2 concentrations > 100 copies/mL, suggesting moderate to severe infection levels in the community.

### 3.2. Wastewater Matrix Effects on Target Gene Amplification

Wastewater matrix effects are a commonly known factor that causes reduced PCR sensitivity. To investigate wastewater matrix effects on the virus amplification, the recovery efficiency of MHV was analyzed alongside water quality parameters for all samples tested. TSS, COD, NH_3_-N, and turbidity (pH was not included due to low variability) correlation matrix from 2700 tests are presented in the supplemental materials, Appendix A. COD, TSS, and turbidity are highly positively correlated with one another as shown by Kendall’s Tau correlation, while NH_3_-N showed low positive correlation with water quality parameters (Appendix A). MHV recovery representing the efficiency of viral concentration, nucleic acid extraction and purification, and multiplex RT-ddPCR steps ranged from 0.13 to 91.2% with an average 27.8% (Figure 4), which is comparable to previous reports [39]. The correlation analysis also showed medium negative correlations (−0.31 to −0.36, *p*-value < 0.05) between MHV recovery rates and TSS, COD, and turbidity. Furthermore, the correlation analysis between wastewater NH_3_-N and MHV recovery revealed that there is no prominent relationship between them (Appendix A), and thus no additional evaluation was conducted with NH_3_-N.

To further evaluate the effects, TSS and COD data were divided into “high” and “low” categories based on their typical values in domestic raw wastewater reported in the literature [40]. “High” TSS was defined as greater than 300 mg/L and anything less was considered “low” TSS. “High” COD was defined as greater than 1000 mg/L and anything less was considered “low” COD. MHV recovery efficiency was compared between the samples with “high” and “low” TSS and COD to evaluate if there was a significant difference between the groups. The results yielded a Mann–Whitney U = 2.91 × 10^4^, Z-score of −9.21, and *p* < 0.05 for TSS, and a Mann–Whitney U = 2.64 × 10^4^, Z-score of −9.96, and *p* < 0.05 for COD, which indicated significant differences between the MHV recovery efficiencies of the “high” and “low” groups. This result suggests that the “low” groups in both instances had a higher recovery efficiency than the “high” groups (negative Z). The sum of ranks (U) was lower for the higher groups, suggesting that higher TSS or COD had lower relative MHV recovery efficiencies. The medians of the lower groups were also higher, further confirming the hypothesis that solids and organics affect viral recovery (Figure 5). This outcome indicates SARS-CoV-2 gene recovery may not be optimal as well and affected by the wastewater matrix.

Due to the clear impacts of wastewater solids and organics on MHV recovery, further testing was warranted to evaluate their effect on the SARS-CoV-2 concentrations. A multiple regression analysis was performed to test the effect of MHV recovery efficiency, solids, or COD on N2 or E gene concentrations. However, the results showed no significant relationship between them. Only 1.9 and 2.7% of variation, for N2 and E genes, respectively, was explained by the variables tested (Appendix A). Additionally, the multiple regression analysis performed on data segregated by high and low groups, designated for TSS and COD and high MHV recovery (>25%) and low MHV recovery (<25%), also did not reveal any significant relationship with variances in N2 and E gene concentrations (Appendix A).

Viral concentration and water quality data are non-parametric in nature, which may affect the determination of statistically significant relationships derived from regression-based analyses. However, the MHV recovery control showed significant associations with water quality parameters such as TSS and COD (Figure 5), as well as variability in recovery efficiencies over the study duration (Figure 4). These results therefore indicate the susceptibility of enveloped coronaviruses, such as SARS-CoV-2 and MHV, to methodological factors and wastewater composition.

## 4. Discussion

This study highlights the importance of thoroughly evaluating metadata and carefully selecting gene targets used for WBS of infectious diseases. There have been efforts to standardize methods of data sharing on public dashboards such as the CDC National Wastewater Surveillance System (NWSS) and WastewaterSCAN, but even these dashboards provide ambiguity in their recommendations or selection process for gene targets for SARS-CoV-2. The CDC, in a now archived webpage, states that primers and probes targeting the N1, N2, and E gene regions have been reported as “specific and sensitive” but also states “when possible, compare wastewater measurements using the same target genes” [41]. The WastewaterSCAN page does not provide recommendations for SARS-CoV-2 primer targets but does mention “using a conserved target on the N gene”, not specifying whether it is the N1 or N2 region [42]. These examples provide just a glimpse into the complexity of target selection and reporting choices when presenting wastewater data for tracking COVID-19. This study took a closer look at the differences in virus concentrations that may have resulted from the choice of a gene target and found the high correlation between the selected targets (N2 and E genes), suggesting that both targets revealed a consistent presence and transmission of COVID-19 in the monitored communities. However, a detailed comparison also indicates that use of the N2 gene target alone would miss a fraction of the positive detection despite the overall agreement between the two gene targets. Contrary to a previous study that suggested excluding E gene target from WBS studies due to its lower frequency of detection than N1 or N2 genes [42], our results show higher detection frequency of E gene than N2. The discrepancies in E gene target detection frequencies between these studies may be attributed to multiple factors, including epidemiological, demographic, and methodological differences.

The primers and gene targets for SARS-CoV-2 detection were developed based on clinical studies of the virus in human specimens and later adopted into WBS efforts. The careful consideration of the benefits and challenges of using a primer target such as the E or N gene for clinical diagnosis is not typically considered on a case-by-case basis in WBS. As stated in a review by Parkins et al., “the selection and diagnostic performance of primer targets are relevant and crucial to accurately tracking and detecting the signal of SARS-CoV-2” [7], yet studies that support primer/probe target considerations seem to have reached an undeserved status quo in the wastewater monitoring field.

The discrepancies in the N2 and E gene results observed in the current study may be due to their structural differences, as well as possible errors incurred through multiplexing. The N gene has a total length of 1260 bp, and the CDC N2 region primer/probe amplicon size is 67 bp, with a guanine–cytosine (GC) of 50.82%. The E gene has a total length of 228 bp, and the WHO E_Sarbeco region primer/probe amplicon size is 125, with a GC of 44.6%. The literature-recommended MHV target has an amplicon size of 108, with a GC content of 41.46% [43]. The higher GC content of the N2 primer indicates that the N2 gene amplicon may require a higher melting temperature for denaturation than the E gene amplicon, possibly affecting the sensitivity and precision of amplification. This amplicon GC content difference is noteworthy when considering the instances of E gene amplification with no N2 amplification reported in this study (Figure 3). The differences in length of the targets for N2, E, and MHV, could have also affected their efficiency of amplification in RT-ddPCR. Given the variabilities in SARS-CoV-2 quantification shown in this study, it is apparent that more careful examination and consideration of primer targets for SARS-CoV-2, and primer targets, in general, for large-scale WBS, are needed.

Wastewater matrix effects remain a significant challenge in wastewater analysis of viruses. As shown in this study, seeded viral recovery is statistically related to the wastewater organics and solids content. Although it is unclear whether the matrix effect impacts different gene targets differently, more investigation is warranted for multiplex primers used in WBS applications, especially primer/probes targeting different gene regions of the same viral genome. Multiplexing is an extreme asset when it comes to affordability and streamlining of RT-qPCR-based quantification. However, the information obtained needs to be carefully analyzed and transparently reported.

Given the distinct differences in SARS-CoV-2 concentrations measured by two different primers in our community-level study, it is apparent that reporting only one gene target may lead to underestimation of true disease prevalence. This study provides supporting perspective for reporting more than one reliable target for SARS-CoV-2, especially when using duplexing or multiplexing primer/probe methods. This may prevent underreporting or even missed signal detection when tracking future outbreaks, especially with COVID-19 endemicity appearing close if not already here. Lastly, our findings contribute to the recommendation of making all viral concentration information available to the public when publishing COVID-19 WBS information, even if a single target’s results are assumed to be more appropriate.

## 5. Conclusions

Ultimately, this study presents unique insights into the research considerations made in a WBS study when using commercially available kits for molecular analysis and the interesting variabilities found in analyzing data for the same viral target. The following conclusions can be drawn from the results of this study:The N2 and E gene targets both have pros and cons when it comes to their specificity and sensitivity to PCR amplification. Results have shown that the E gene had a higher efficiency of amplification during particularly low viral activity periods.PCR inhibition is an important challenge in WBS studies. Matrix recovery controls, such as MHV, can be a useful indicator of inhibition in wastewater monitoring studies. MHV recovery efficiency in this study clearly indicated inhibition in samples with higher TSS and COD, even with careful removal of debris in wastewater samples and purification of RNA.WBS programs require reproducibility and cohesion for success and deployment of sustainable, passive surveillance networks. More in-depth methodological studies should be conducted to help standardize the approach to wastewater sample collection, pathogen detection and/or quantification, and data sharing.

## Figures and Tables

**Figure 1 microorganisms-13-01862-f001:**
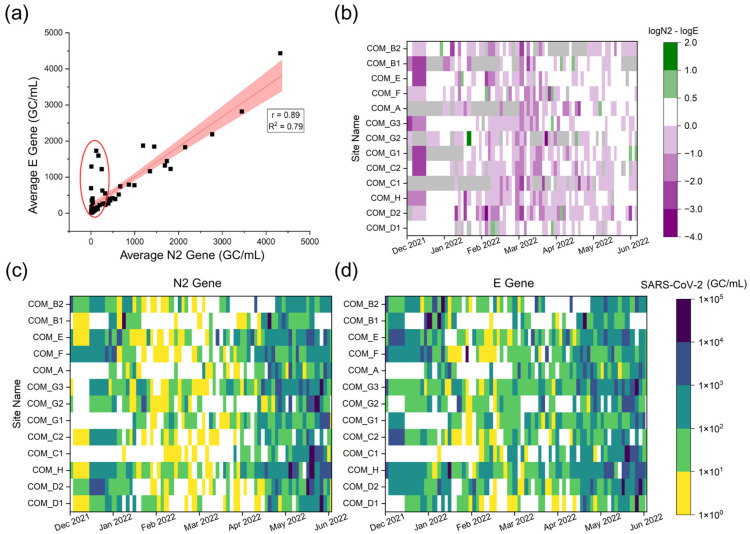
Comparison of N2 and E gene concentrations (GC/mL) among all samples. (**a**) A linear regression between the N2 and E gene concentrations for the entire study, with the anomalies of interest circled in red. (**b**) The heat map of SARS-CoV-2 concentration differences determined by N2 or E gene at each manhole site (presented as logN2-logE). Gray shading represents a sample for which the calculation could not be performed due to missing values. (**c**) SARS-CoV-2 concentrations determined by N2 gene, and (**d**) SARS-CoV-2 concentrations determined by E gene.

**Figure 2 microorganisms-13-01862-f002:**
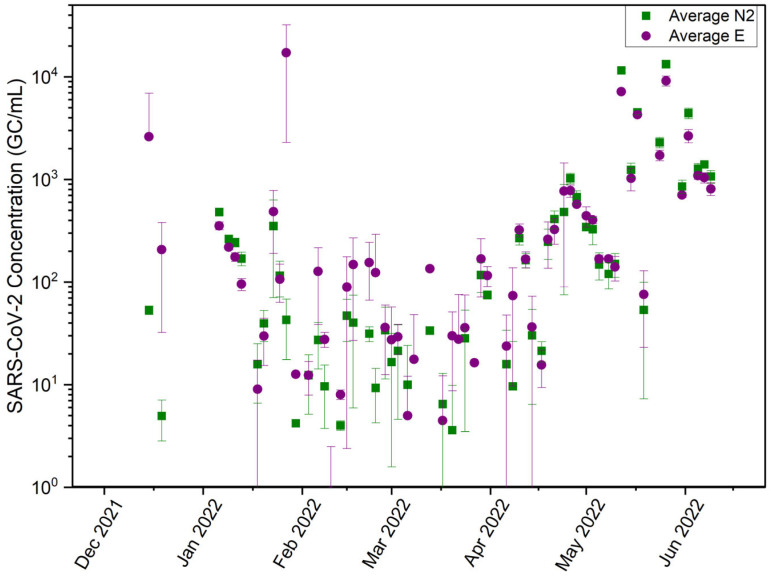
SARS-CoV-2 gene concentrations determined by N2 or E gene (GC/mL) at manhole COM_H.

**Figure 3 microorganisms-13-01862-f003:**
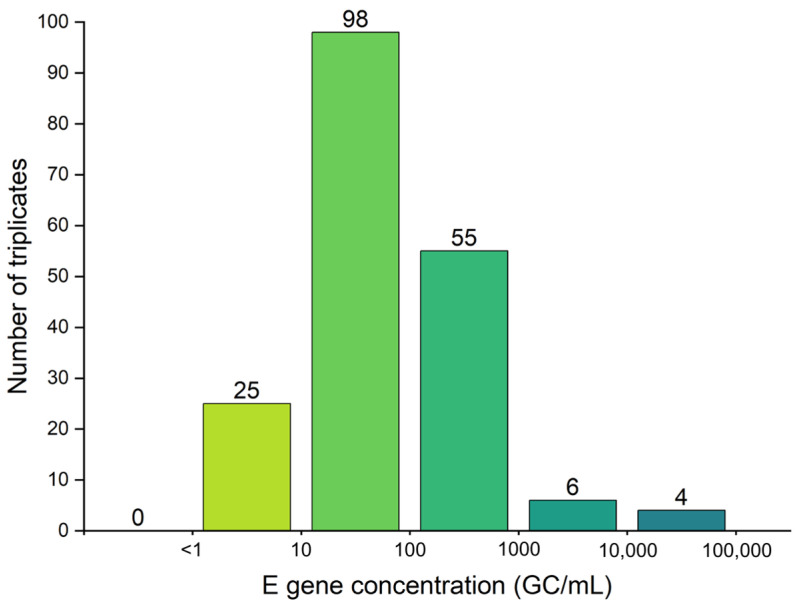
E gene concentrations of the triplicates when the N2 gene was “non-detect” = 0 GC/mL in samples (n = 188 out of total 2121 triplicates).

**Figure 4 microorganisms-13-01862-f004:**
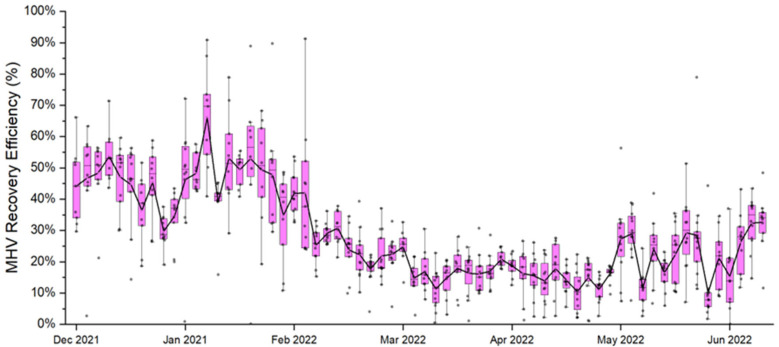
Average MHV recovery efficiency (%) for all samples analyzed for SARS-CoV-2. Data points for each date represent all sites, 13 in total.

**Figure 5 microorganisms-13-01862-f005:**
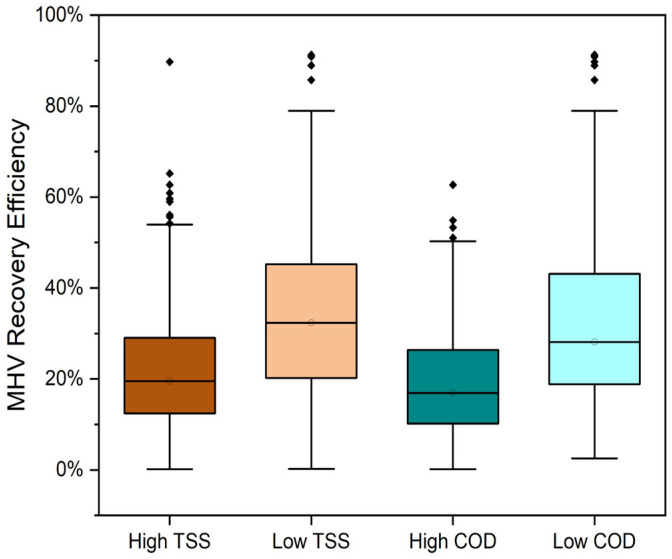
Comparison of high and low TSS and COD on the virus recovery. All *p*-values < 0.05.

## Data Availability

The original data presented in the study are openly available in Mendeley Data at doi: 10.17632/6x5zffpxwb.3.

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
