# Peer review of "Variabilities in N2 and E Gene Concentrations in a SARS-CoV-2 Wastewater Multiplex Assay"

_microorganisms, 2025, doi:10.3390/microorganisms13081862_

Round 1

Reviewer 1 Report

Comments and Suggestions for Authors

The authors have presented interesting results of a well planned and executed study on covid-19 detection in wastewater. There are only a few comments and questions in the attached commented manuscript that should be considered before the paper is published.

Author Response

Response to Reviewer 1:

Thank you very much for taking the time to review this manuscript; we appreciate your thoughtful insights. Please find the detailed responses below and the corresponding revisions/corrections highlighted/in track changes in the re-submitted files.

2. Questions for General Evaluation

Reviewer’s Evaluation

Response and Revisions

Does the introduction provide sufficient background and include all relevant references?

Yes

Agree with all the evaluations provided by Reviewer 1.

Are all the cited references relevant to the research?

Yes

Is the research design appropriate?

Yes

Are the methods adequately described?

Yes

Are the results clearly presented?

Yes

Are the conclusions supported by the results?

Yes

3. Point-by-point response to Comments and Suggestions for Authors

Comments 1: “Choose keywords that don´t appear in the title.”

Response 1: Thank you for pointing this out. We agree with this comment. Therefore, we have updated the keywords. Changes can be found in the revised manuscript (Page 1, Line 32-33).

“Keywords: Wastewater-based surveillance; manhole; COVID-19; RT-ddPCR; primer; recovery efficiency”

Comments 2: “Do the authors think other target sequences could/should be considered also?”

Response 2: This is a good question. Other SARS-CoV-2 primer targets are mentioned in the introduction, but we agree further investigation is warranted to understand the variability of other primer targets and their associated viral concentrations in wastewater. We have, accordingly, revised this sentence to improve the clarity of the focus of this study and broader implications (Page 3, Paragraph 2, Lines 110-111).
“… this study hopes to better understand 1) the best approach for reporting SARS-CoV-2 concentrations and 2) which physio-chemical characteristics of wastewater may have contributed to the over or under-amplification of the gene targets during RT-ddPCR.”

Comments 3: “Freezing samples can alter their physicochemical characteristics. The authors should comment on this.”

Response 3: Thank you for pointing this out, we completely agree, and have revised this sentence to make the impact on freeze/thaw cycles on sample integrity clearer (Page 3, Paragraph 4, Lines 129-1230).

“Wastewater aliquots were stored at -80°C upon arrival to the lab for 2 to 12 months, until all the samples were processed. Before testing, frozen samples were thawed at 4°C slowly for 24 to 48 hours to minimize interference with sample integrity, and subsequently analyzed for COD, TSS, NH3-N, pH, and turbidity.”

Comments 4: “This unit should be written out extensively “genome copies" -  since it is not comm and may not be understood by the less experienced reader.”

Response 4: Thank you for pointing this out, we completely agree, and have added the full definition of the acronym before it appears in text. (Page 4, Paragraph 2, Line 157).

“… and MHV are 5, 2, and 30 genome copies (GC)/ μL, respectively.”

Comments 5: “Why do the authors believe that 71 cases are not also relevant?”

Response 5: Thank you for providing this comment. We agree that the wording of this sentence indicates a meaning that does not reflect the significance of these instances. We have changed the wording accordingly (Page 6, Paragraph 1, Line 227).

“In contrast, there were 71 instances…”

Comments 6: “comma in wrong position, should be placed after "effects"

Response 6: Thank you for providing this comment. We agree that the grammar was incorrect, and we have edited the sentence accordingly (Page 8, Paragraph 1, Line 257).

“To further evaluate the effects, TSS and COD data were divided…”

Comments 7: “Very high value for domestic sewage, no?

Response 7: Thank you for providing this comment. We agree that it is a high value but this was the intent, and there are two main considerations that supported this categorization: 1) we were sampling at the sub-community level from sewer manholes, resulting in highly concentrated wastewater samples, upstream of treatment plants, with higher than average COD levels and 2) ‘Introduction to Environmental Engineering’ (Davis and Cornwell, 2013) provides the typical composition for untreated domestic wastewater in Table 8.1 and lists “Strong” COD at 1000 mg/L or higher.

No changes were made to the text but let us know if you still have concerns regarding this categorization.

Comments 8: “Belongs in Methods.”

Response 8: Thank you, this is a good point. This has been addressed accordingly with a sentence being moved to the “Data Analysis” section of methods and redacting of details in the results.

Part staying in Results: (Page 8, Paragraph 1, Lines 261-263)

“MHV recovery efficiency was compared between the samples with “high” and “low” TSS and COD to evaluate if there was a significant difference between the groups. The results…”

Part moved to Methods, 2.3 Data Analysis: (Page 4, Paragraph 3, Line 166-168)

“Mann-Whitney U Test, a non-parametric statistical test, was performed to compare the MHV recovery efficiency between samples categorized into different water quality groups.”

Comments 9: “GC content?”

Response 9: Thank you for providing this comment, as it made it clear that there was confusion between “genome copies” and the guanine-cytosine (GC) content, which is a helpful genomic composition parameter. In the text, I have defined GC content before having the acronym appear (Page 9, Paragraph 3, Line 330).

 “…is 67 bp, with a guanine-cytosine (GC) content of 50.82%.”

Comments 10: “...higher GC content of the N2 primer indicates...”

Response 10: Thank you for your comment, we agree with your suggestion and have edited the wording accordingly (Page 9, Paragraph 3, Line 333).

The higher GC content of the N2 primer indicates…”

Comments 11: “How can this account for the cases where N2 was detected, but not E?”

Response 11: Thank you for pointing this out as it provides a critical perspective that was missed when writing this sentence. This sentence has been removed since it was not helpful to the discussion but rather, convoluting (Page 9, Paragraph 3, Lines 338-340).

Sentence removed.

Comments 12: “Was total or soluble COD quantified? Organics might not be the problem. It would be interesting to know if the TSS were largely organic (VSS).”

Response 12: Thank you for bringing this up. In this study, total COD was tested using the HACH test kit described in the methods. Unfortunately, we did not measure VSS when performing our TSS test, but this is a great suggestion for the future.

As mentioned, there was a high correlation between TSS and COD shown through the Kendall’s Tau test in the supplementary information, so this leads us to hypothesize that many of solids were organic material.

No changes have been made to the original text.

4. Response to Comments on the Quality of English Language

Point 1: N/A

Response 1:   N/A

5. Additional clarifications

No additional clarification! Thank you for your considerate review of our manuscript. Hopefully my revisions have sufficiently addressed your comments.

Reviewer 2 Report

Comments and Suggestions for Authors

In the manuscript by Green et al., the authors evaluate the sensitivity of N2 and E gene primer sets for detection of SARS-CoV2 RNA in wastewater samples. Authors find a direct correlation between N2 and E gene concentrations in samples with higher viral loads, however, at low viral loads the E gene is more readily detected than the N2 gene. These results suggest that quantifying only one of the genes can give misleading results, specifically the preference of using the N2 gene as a target, which is not detected at low concentrations and could suggest lack of virus in samples when in reality there is just an inability to detect the gene. Authors also investigated whether viral recovery efficiency and water quality had an effect on gene detection but could not find a clear relationship. Water quality had an effect on virus recovery efficiency probably due to PCR inhibition and this should be taken into account when standardizing the method.

The manuscript has a clear aim which is accomplished. Introduction covers the why the study should be done. Methods and results are clearly stated and discussion puts the results into perspective.

I recommend accepting the manuscript.

Author Response

Thank you very much for taking the time to review this manuscript; we appreciate your thoughtful insights.